

# An emulation-based approach for interrogating reactive transport models

**Authors:** Angus Fotherby[1*], Harold J. Bradbury[1], Jennifer L. Druhan[2], Alexandra V. Turchyn[1]
1: Department of Earth Sciences, University of Cambridge, Cambridge, UK
2: Department of Geology, University of Illinois at Urbana Champaign, Urbana, IL
*corresponding author: af606@cam.ac.uk
**Keywords:** Environmental remediation, Reactive transport, Machine learning, Sensitivity analysis
**Abstract**
We present a new approach to understand the interactions among different chemical and biological
processes modelled in environmental reactive transport models (RTMs) and explore how the
parameterisation of these processes influences the results of multi-component RTMs. We utilize a
previously published RTM consisting of 20 primary species, 20 secondary complexes, 17 mineral
reactions and 2 biologically-mediated reactions which describes bio-stimulation using sediment from
a contaminated aquifer. We choose a subset of the input parameters to vary over a range of values.
The result is the construction of a new dataset that describes the model behaviour over a range of
environmental conditions. Using this dataset to train a statistical model creates an emulator of the
underlying RTM. This is a condensed representation of the original RTM that facilitates rapid
exploration of a broad range of environmental conditions and sensitivities. As an illustration of this
approach, we use the emulator to explore how varying the boundary conditions in the RTM
describing the aquifer impacts the rates and volumes of mineral precipitation. A key result of this



work is the recognition of an unanticipated dependency of pyrite precipitation on pCO₂ in the
injection fluid due to the stoichiometry of the microbially-mediated sulphate reduction reaction. This
complex relationship was made apparent by the emulator, while the underlying RTM was not
specifically constructed to create such a feedback. We argue that this emulation approach to
sensitivity analysis for RTMs may be useful in discovering such new coupled sensitives in
geochemical systems and for designing experiments to optimise environmental remediation. Finally,
we demonstrate that this approach can maximise specific mineral precipitation or dissolution
reactions by using the emulator to find local maxima, which can be widely applied in environmental
systems.
**Synopsis**
This study explores key factors regulating mineralization reactions in near surface environments
revealed by a machine learning approach to reactive transport modelling.
**1    Introduction**
Reactive transport modelling has been extensively applied across a wide variety of environmental
systems, providing a powerful means of quantifying, and even predicting, processes across Earth's
(near) surface environments (Richter and DePaolo, 1987; Bain et al., 2000; Johnson et al., 2004; van
Breukelen et al., 2004; Gaus et al., 2005; Torres et al., 2015; Li et al., 2017; Arora et al., 2020;
Molins and Knabner, 2020; Rolle and Borgne, 2020; Druhan et al., 2020; Cama et al., 2020).
Reactive transport models (RTMs) are constructed by combining multiple physical, chemical, and
biological processes to simulate the behaviour of environmental systems. As applications and
software have concurrently expanded (Steefel et al., 2015; Li et al. 2017; Maher & Mayer, 2019;
Druhan & Tournassatt, 2019), it is becoming increasingly common to explicitly calculate the rates of
production and consumption for a variety of coexisting chemical species, as well as their equilibria



with mineral phases, and their transport as they evolve in time and space. This type of multi-phase,
multi-component RTM is a type of forward modelling where the results of the simulation emerge
from a complex suite of interacting pathways, and hence the causes of observed behaviour are not
always obvious.
RTMs are often designed to describe the behaviour of specific field sites and systems. Due to their
process-based nature, designing RTMs requires selection of a suite of chemical reactions and
transport mechanisms which are thought to dominate the geochemistry of the system over the scales
of interest. However, the parameterisation of various selected processes is often not unique and can
impact system behaviour (Williams et al., 2011; Martinez et al., 2014; Seigneur et al., 2021; Steefel
et al., 2005a). To assess the impact of the choice of parameterisation and the values chosen for
different parameters on model predictions, sensitivity analyses are generally performed (Malaguerra
et al., 2013; Gatel et al., 2019). However, as RTMs become increasingly sophisticated, they
incorporate disparate processes that can interact with each other in complex ways (Dwivedi et al.,
2018; Hubbard et al., 2018, 2019; Maavara et al., 2021a, b; Dwivedi et al., 2017).
The sensitivity analysis of an RTM in application to a specific environmental system can elucidate
the relative importance of specific interactions; for example, testing the solubility of mineral phases
relative to changes in the solution chemistry. However, results might emerge that were not
anticipated. These results might represent a real, but unexpected, interaction in which case the
sensitivity analysis has yielded new insight into the system being modelled. Equally, the result might
represent an incorrect interaction between two different processes that are known to act
independently of each other, in which case the RTM can be improved. Unfortunately, due to the
computational expense of many modern multi-component RTMs (e.g. Abd and Abushaikha, 2021;
Seigneur et al., 2021; Gharasoo et al., 2022), it is normally impractical to perform sensitivity analyses
in more than a few dimensions and it is up to the investigator to use their knowledge of the system to



choose which sensitivity analyses are necessary to explore (Steefel et al., 2005b). Ideally, we would
be able to systematically perform sensitivity analyses over many model parameters, considering how
model outputs vary as a function of multiple input parameters simultaneously (i.e. in a multivariate
way), while also lightening the computational burden that commonly occurs when using inverse
modelling approaches implemented by codes like PEST and iTOUGH2 (Doherty, 2004; Finsterle et
al., 2017). Such a capacity could direct future laboratory-based investigations to test whether these
model results are real-world phenomena, ultimately offering improved parameterisation of critical
components within the reaction network.
Here, we demonstrate an approach to explore a wide variety of potential model parameters, by
adapting an emulation method similar to that previously applied in physics-based animation
(Grzeszczuk et al., 1998) to complex multi-physics simulators (Lu et al., 2021; Bianchi et al., 2016)
and climate models (Beucler et al., 2019; Krasnopolsky et al., 2005; Castruccio et al., 2014;
Kashinath et al., 2021) as well as applied to emulating fluid flow through Dolomite using a neural
network (Li et al., 2022). In this emulation approach, the underlying physical system is approximated
by a statistical model (the emulator) which can be evaluated more quickly than a conventional
forward model. How this emulator is constructed varies by implementation and may encode
assumptions about the underlying system to be modelled (e.g. conservation of energy (Beucler et al.,
2019)). In our implementation the emulator is built by training a Gradient Boosted Trees (GBT)
regressor (Chen and He, 2015) on a synthetic dataset generated from the original RTM. By training
such a GBT model on the synthetic dataset generated by the original RTM, we create an emulator of
the original system. This emulation approach is general and can be applied to a wide range of RTMs,
using "off the shelf" statistical libraries, requiring no special construction of the statistical model
beyond the choice of some training parameters. This approach can identify the critical processes and





parameters within RTMs and address the requirement for comprehensive, multivariate sensitivity
analyses.
We first present a tool that automates creation of synthetic datasets: a Python wrapper for the RTM
software CrunchTope (Druhan et al., 2013; Steefel et al., 2015), which we have named Omphalos.
Omphalos edits and runs CrunchTope input files in an automated fashion, systematically changing
model parameters according to user specification. It then records the output data, along with the
corresponding model input parameters for later analysis. We then apply a machine learning method
(Gradient Boosted Trees) to these recorded inputs and outputs to create a predicative model that can
reproduce RTM outputs based on the input variables, which we term a Reactivate Transport
Emulator (RTE).
We envision that such Reactive Transport Emulators could be used to direct new experimental
investigation to identify and corroborate predicted dependencies; providing much-needed
multivariate analysis of RTMs and helping to identify effects that can, in the future, be considered
explicitly when developing new RTMs. In pursuit of this goal, we demonstrate our emulator
approach in application to an RTM built for biostimulation of a contaminated aquifer. We also show
an additional application of this approach to efficiently predict the condition which maximises an
RTM-predicted time-integrated rate over the set of chosen parameters. We also present, in the
Supporting Information, another example in application to a deep-sea sediment column.
**2      Description of the Case Study**
**2.1   Old Rifle Site, Colorado**
The Old Rifle site is located near Rifle, Colorado, USA. The location historically hosted a vanadium
and uranium ore processing facility, and the groundwater at the site remains high in aqueous



uranium. Oxidised uranium (U(VI)) is fluid-mobile and highly toxic, while reduced uranium (U(IV))
is much less soluble and forms stable precipitates such as uraninite ($UO_2$) (Anderson et al., 2003; Wu
et al., 2006; Dullies et al., 2010; Williams et al., 2011; Long et al., 2015). Thus, uranium reduction
has been suggested as a means for remediating uranium contamination in groundwater. It has been
shown that iron sulfide minerals ($FeS_{2(s)}$) aid the reduction of soluble U(VI) to insoluble U(IV)
precipitates even after active remediation has ceased (Komlos et al., 2008; Moon et al., 2010; Bargar
et al., 2013; Long et al., 2015; Bone et al., 2017).
The RTM published for Old Rifle, upon which the RTE is based, was originally created as a
comprehensive model of microbial sulfate reduction and sulfide precipitation in Old Rifle sediment
during stimulation of microbial activity by amendment with $C_2H_3O_2^-$ (Druhan et al., 2014) (for a
schematic illustration of this RTM, see Fig. S2). In this context, we choose to vary the influent
boundary condition chemistry, representing changes to the chemical composition of the artificial
groundwater injectate. The original experiment was designed to model microbial sulfate and iron
reduction in the sediment; therefore, we use net amorphous iron (II) sulfide ($FeS_{(am)}$), and pyrite
($FeS_{2(s)}$) precipitation (both hereafter referred to simply as 'pyrite') as an observable that will record
the sensitivities of the model predictions to changes in the injection fluid. We also demonstrate the
utility of the emulator in predicting the chemical composition of the injection fluid that will
maximise the volume of pyrite precipitated in the sediments when amended with a labile organic
carbon source via injection wells.
**3    Methodology**
**3.1    General Strategy**
To explore the dependence of the RTM on the chosen environmental variables, we begin with a
Monte Carlo approach; we draw random values for each parameter and record the model output



under that randomised condition. We then fit a model to this Monte-Carlo-generated dataset using a
GBT regressor. This fitting results in a model (our emulator—RTE) that reproduces the complex
interdependencies of chemical species that are encoded in the original, underlying, RTM. This
emulator can be interrogated to examine the dependence of the RTM outputs on the originally chosen
environmental variables in an efficient, multivariate way. This new way of performing sensitivity
analyses has the potential to give insight into trends and relationships that would not be apparent
otherwise and ultimately allows us to investigate the sensitivity of the model outputs with respect to
the RTM's original parameterisation. First, we will describe how we use the Monte Carlo approach
to generate data and then how we fit a model to this data. The overall workflow is shown in Figure 1.






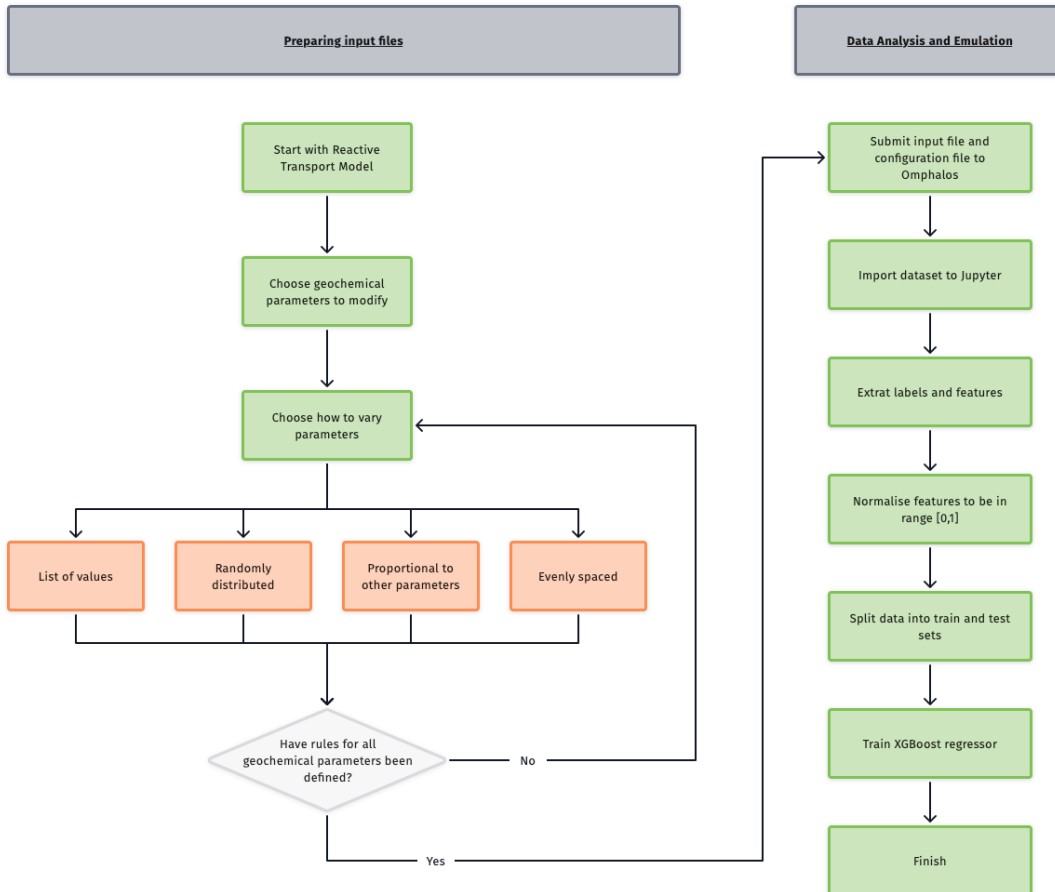

**Figure 1: Flowchart describing the overall reactive transport emulation workflow developed in this study. It is divided into two key sections: preparation of the input reactive transport model for submission to Omphalos, and the analysis and emulation of the resultant data.**

**3.2   Generating Data**

We use the open-source software CrunchTope as the reactive transport framework for the models in

this study. To generate the synthetic datasets necessary for our approach and given the time-

consuming nature of generating a single point (requiring a complete run of the RTM, along with

modified boundary conditions), we developed a software package in Python to automate this process.

This software package can manage the automatic generation and submission of unique input files to





CrunchTope, as well as recording the output of each run, storing it in a manageable data structure for
future use. Use of the software package is straight-forward, requiring the configuration of a single file
listing which species/parameters are to be varied, and how they should be varied.
We have named this software package Omphalos (available for download—Sect. 6.1). Omphalos can
be run on clusters using Simple Linux Utility for Resource Management (Yoo et al., 2003) to execute
input files in parallel, which considerably reduces the time required to generate large datasets.
Omphalos works by taking random values which are drawn from uniform distributions (other
statistical distributions are possible) of the chosen variables, sampling the space evenly. This
provides a complete dataset for training the emulator.
While the underlying principle of training emulators on synthetic data can be applied to any reactive
transport code, currently the software used to implement the approach is only compatible with
CrunchTope, because the input file reading and writing must be in a specific format. The approach is
readily generalized, however, and the methodology could be applied to any RTM software (e.g.
Geochemist's Workbench, ToughReact), provided that the string input/output code is adapted for
compatibility. To use other RTMs with Omphalos, two key factors need to be addressed:
compatibility with Omphalos, and the computational expense of a single RTM run.
**3.3    Application to Contaminated Aquifer Case Study**
We begin by applying the emulation methodology to our case study. To create the dataset for training
the emulator, we collected the results of 10,927 unique CrunchTope simulations based on the original
RTM describing Old Rifle using Omphalos, drawing random concentrations for each species in the
boundary condition. Of these 10,927 runs, 9416 provide useable data because some runs fail to
converge within the specified timeframe, or the geochemical condition generated cannot be charge
balanced by CrunchTope. The concentrations for $NH_4^+$, $SO_4^{2-}$, $Ca^{2+}$, and $C_2H_3O_2^-$ are varied between



0–30 mM. The pCO2 is varied between 0–10 bar. We acknowledge that these ranges of
concentrations are somewhat higher than those that occur in natural systems, but we extend the range
to observe RTM behaviours at limiting concentrations. Related to this, it is possible for the dominant
reaction mechanism in a system to change under differing conditions (e.g. the change in calcite
dissolution mechanism as a function of pH (Dolgaleva et al., 2005)) and any such behaviour should
be explicitly encoded into the RTM, otherwise the emulator may give invalid predictions under
conditions that are far from the original model run. We have assumed in this study that the
mechanisms governing the precipitation of pyrite do not change under very low or very high
concentrations of these species.
The injection fluid was constrained at pH 7.2. This constraint, in conjunction with the concentration
of various species iterated in Omphalos, speciates according to CrunchTope's internal speciation
calculation. Therefore, for example, although the total amount of $SO_4^{2-}$ in the injection will be iterated
in, and dictated by, Omphalos, the amount that speciates into other aqueous complexes (i.e.
secondary species) like $HSO_4^-$ or $H_2SO_{4(aq)}$ is controlled by CrunchTope. For the sake of simplicity,
we will report the input concentration, not the concentration after speciation.
The RTM describing Old Rifle has 100 grid cells with a size of 1 cm. Each run of the RTM took
approximately 90 seconds, so the total time to generate the dataset was roughly four hours when run
on a remote machine with 200 CPUs. The number of runs was chosen as a balance between what was
computationally tractable and the ability of the emulator to achieve a good fit. We have intentionally
chosen to vary some chemical species in the influent boundary condition that do not play an obvious
role in the mineral precipitation process we are particularly interested in, namely, the precipitation of
pyrite in Old Rifle sediments (e.g., $NH_4^+$ or $Ca^{2+}$, respectively). We did this to see if we can use the
emulator to detect behaviour in the RTM beyond what we might initially hypothesise.



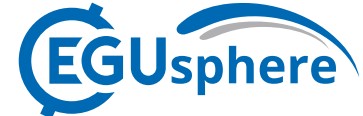

### 3.4 Fitting the emulator


We implement the GBT regressor using XGBoost (Chen and He, 2015) in Python. The code for
fitting the models is available in the Supporting Information. For a precis on GBT models, see the
supplement Sect. S1.2.

### 3.4.1 Data Strategy


Data generated by Omphalos was imported into a Jupyter notebook environment from the .pkl output
file. There are 9416 different input file runs in this data file. The relevant data was indexed out of the
data structure; in our case this meant the concentrations of $NH_4^+$, $SO_4^{2-}$, $Ca^{2+}$, and $C_2H_3O_2^-$ in the
boundary condition, as well as value of pCO2. This results in a 5x9416 array of floating-point
numbers for the features. Each feature was then normalised to be in the range 0 to 1 for training. For
example, values of $SO_4^{2-}$ concentration in the simulations were drawn randomly between 0 and 30
mM, so all $SO_4^{2-}$ concentrations were divided through by 30 to have values in the range 0–1. We did
this to improve the training performance of the GBT model over different datasets (i.e. so that the
same GBT model can be applied to both the Old Rifle case study, and our supplementary case study
of ODP Site 1086 (see Supplement, Sect. 3).
Similarly, the relevant data was also extracted from the data file: for each cell in the gridded RTM,
we calculated the net pyrite precipitation over the course of the simulation, and then summed this
value over the column to get the net pyrite precipitated across the domain. This results in a 1x9416
array of floating-point labels to be predicted from the feature array. We scale this feature array by a
factor of $1x10^4$ to avoid issues with small floating-point numbers in XGBoost.
We prepared these data for training the GBT regressor with a hold-out strategy using the
scikitlearn.train_test_split method, keeping 10% of the dataset back for validating the model. Data
was split randomly within the dataset. This means that 8474 randomly selected data points were used





to train the model and 942 randomly selected data points were used to test it by using the model to
predict a value based on the held back data and comparing the prediction to the true value.
**3.4.2 Training Strategy**
We use the test set of data points generated by Omphalos to train an XGBoost regressor using
squared error as the loss function to predict the amount of pyrite precipitated in the column as a
function of varied species concentrations in the boundary condition. Squared log loss, and pseudo-
Huber error we also tried but squared loss performed best overall. Training curves showing the
testing and training loss as training progressions are given the supplement, Fig. S4.
Hyperparameter choices for the model are explained and given in the supplement, Sect. S1.3, Table
S1. The choice of hyperparameters is the same for each emulator model, and we are able to achieve
high quality fits using the default XGBoost regularisations, only changing a few settings relating to
tree growth policy. While it is a known problem in machine learning that the choice of optimal
hyperparameter is dependent on the data being modelled (Claesen and De Moor, 2015), it appears
that in the context of these RTEs, the hyperparameters chosen give a good fit for both Old Rifle and
our supplementary case study of ODP Site 1086: datasets describing very different natural
environments, with different length and time scales. This makes the workflow applicable across a
wide variety of reactive transport modelling domains.
It is possible that with more complex hyperparameter tuning, better emulator fits may be achieved,
but for the purposes outline in this paper, we suggest that this automated optimisation of a subset of
the available hyperparameters is sufficient, and represents a balance between emulator fit,
generalisability across differing RTMs, and time spent by the user.
**4      Results and discussion**
**4.1    Application to the Old Rifle Site**




The synthetic data generated using Omphalos to interrogate the underlying RTM are shown in Fig. 2,
colour mapped by the $pCO_2$ with which the injectate solution is in equilibrium. The colour mapping
helps visualise how variability in the precipitated volume of pyrite over the 43-day RTM simulation
might be considered in conjunction with other model parameters. Ultimately, pyrite forms because
aqueous hydrogen sulfide, produced through microbial sulfate reduction, reacts with reduced ferrous
iron (Fe(II)) to form pyrite. Thus, we aim to explore the interdependencies between the mechanisms
driving microbial sulfate reduction and the subsequent precipitation of pyrite, as they emerge due to
variations in injectate chemical composition.





**Figure 2: Scatter plots of chemical concentrations in the fluid injectate (influent boundary condition) for an RTM adapted to Old Rifle sediments colour-mapped by the pCO₂ with which the inlet boundary condition is in equilibrium. The dataset comprises 9416 points generated by drawing concentrations for all five species independently from uniform random distributions, with the corresponding net increase in pyrite volume fraction precipitated (y-axis) calculated by running the Old Rifle RTM designed by Druhan et al. (2014) with the randomised influent**



**boundary condition. The green diamond indicates the net pyrite volume fraction generated**
**from the original boundary condition used in Druhan et al. (2014).**
We then train the emulator on this synthetic dataset. Fitting a GBT regressor to the data in Fig. 2
means Fig. 3 can be generated by the emulator. This figure shows how the emulator predicts the
change in pyrite volume fraction as the concentration of each of the species in the injection fluid is
varied (other species in the RTM not defined as variables in this study are held constant at values
reported by Druhan et al. (2014)). The convergence of the emulator is shown in Fig. S3. We stress
that the RTM results shown in Fig. 3 are not part of the training dataset, and that the emulator has not
been exposed to these exact values. This demonstrates the capability of the emulator to reproduce the
underlying RTM itself. For example, Fig. 2A suggests visually that the concentration of $NH_4^+$ in the
system is uncorrelated with net pyrite precipitation at the Old Rifle Site. Fig. 3A confirms this lack of
dependence on $NH_4^+$.
In contrast to the minimal impact that changing $NH_4^+$ concentration has on pyrite precipitation,
$C_2H_3O_2^-$ and $SO_4^{2-}$ concentrations correlate strongly with net pyrite precipitation. This is as expected
in a system where $C_2H_3O_2^-$, which is the electron donor for microbial sulfate reduction, enables
sulfate to be reduced to sulfide and thus drive pyrite precipitation in the presence of Fe(II).
Approximately 20 days after $C_2H_3O_2^-$ amendment, microbial sulfate reduction takes over from
dissimilatory iron reduction as the dominant process consuming $C_2H_3O_2^-$. As microbial sulfate
reduction requires eight-times the number of electrons per mole of $SO_4^{2-}$ reduced than dissimilatory
iron reduction requires (per mole of iron reduced), the electron donor ($C_2H_3O_2^-$) begins to be rapidly
consumed, whereas during dissimilatory iron reduction it was effectively in excess. As a result of this
new scarcity of $C_2H_3O_2^-$, the rate of dissimilatory iron reduction drops and so does the concentration
of Fe(II). However, dissimilatory iron reduction is still active in the column, releasing a small—but




non-zero—flux of aqueous Fe(II) that allows for continued pyrite precipitation. The emulator
interprets this as Fe(II) being 'always' available in this system, and thus predicts that pyrite
precipitation can scale linearly with $SO_4^{2-}$ and $C_2H_3O_2^-$, as shown in Fig, 4A. The sediment itself
would need to contain abundant ferrihydrite, goethite, or another bioavailable ferri(hydr)oxide for
this reduction to continue indefinitely; this may not be the case. This highlights the need for the range
of parameters sampled when training the emulator to be sufficiently wide to capture all the RTM
behaviour, otherwise it may extrapolate and "learn" incorrect assumptions about the system: in this
case that bioavailable iron never limits dissimilatory iron reduction. One solution would be to expand
the range over which concentrations are drawn to reach the limit where iron-bearing mineral volume
fraction becomes a limiting factor so that the model can learn what happens when this occurs.

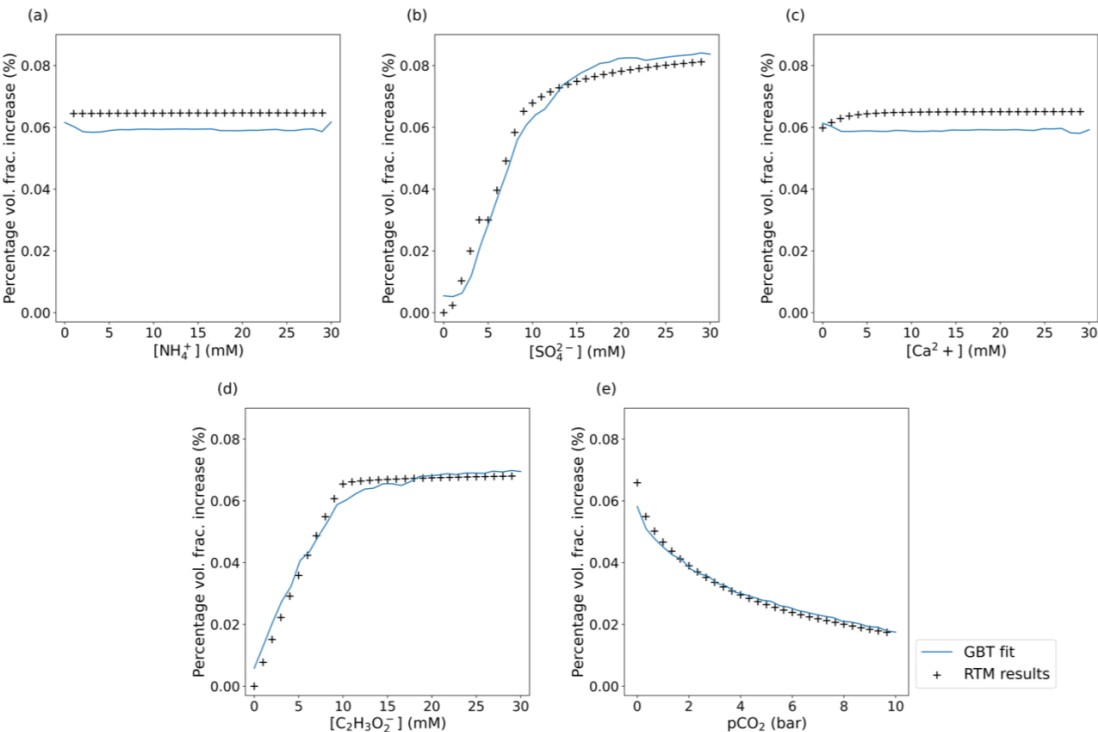


**Figure 3: Plots of the GBT model fit (blue line) plotted over the results from the underlying**
**RTM (black + symbols) when interrogated with the same input parameters (which are taken as**



**ground truth). Each plot shows the net volume fraction due to pyrite precipitation as a**
**percentage of the initial volume fraction of the sediment as each parameter is varied while all**
**other parameters are held at the values used in the original experiment by Druhan et al. (2014).**
**The emulator (blue line) captures the overall trends in the data. The lack of smoothness in the**
**emulator predications arises from the inability to encode this as a condition in XGBoost and**
**the discreet nature of the decision tree algorithm.**
We also note that our emulator suggests that increasing $pCO_2$ leads to decreased pyrite precipitation
(Figure 4E), a relationship that may not have been apparent in a single run of the RTM. Three-
dimensional visualisation of the data confirms that the pyrite-volume-fraction-change varies as a
function of $pCO_2$ net pyrite precipitated decreasing as $pCO_2$ increases (Fig. 4B and Fig. 4C). This
three-dimensional visualisation allows us to see that the gradient of the pyrite-volume-fraction-
change with respect to $SO_4^{2-}$ and $C_2H_3O_2^-$ is itself a function of $pCO_2$ and flattens as $pCO_2$ increases.
To understand why the gradient changes, we must first understand why $pCO_2$ affects the amount of
pyrite precipitated in the first place.
Sediment samples from Old Rifle are initially poised for dissimilatory iron reduction and there is a
sizeable community of iron-reducing bacteria naturally present in the system. The background
sulfate-reducing microbial community is initially relatively small and thus, for microbial sulfate
reduction to proceed at significant rates, the mass of sulfate-reducing bacteria must first increase. In
the original experiment by Druhan et al. (2014), the sulfate-reducing biomass begins reaching a size
where it can start consuming large quantities of $C_2H_3O_2^-$ around day 20 of the experiment. This
biomass growth is modelled in CrunchTope using a Monod-biomass rate law (Jin and Bethke, 2005),
which has both an anabolic and catabolic component. In the formulation of this Monod-Biomass rate
law as implemented in CrunchTope, the thermodynamic term (Gibbs free energy of the reaction) is
calculated exclusively using the catabolic pathway. The catabolic pathway for this reaction (in terms



of the exchange of one electron) is given below in Equation (4.1), and the form of the Gibbs free
energy is this context is given in Equation (4.2) (we take the phosphorylation potential to be 0, and
the average stoichiometric number to be 1, see derivation in Jin and Bethke (Jin and Bethke, 2005)
for further details).
$$\tfrac{1}{8}\mathrm{SO_4^{2-}} + \tfrac{1}{8}\mathrm{C_2H_3O_2^-} + \tfrac{3}{8}\mathrm{H^+} \rightarrow \tfrac{1}{8}\mathrm{H_2S_{(aq)}} + \tfrac{1}{4}\mathrm{CO_{2(aq)}} + \tfrac{1}{4}\mathrm{H_2O} \tag{4.1}$$

$$\Delta G = \mathcal{R}\mathcal{T}\ln\left(\frac{[\mathrm{CO_{2(aq)}}]^{\frac{1}{4}}[\mathrm{H_2S_{(aq)}}]^{\frac{1}{8}}}{[\mathrm{SO_4^{2-}}]^{\frac{1}{8}}[\mathrm{C_2H_3O_2^-}]^{\frac{1}{8}}[\mathrm{H^+}]^{\frac{3}{8}}}\right) \tag{4.2}$$

Taking this form for the Gibbs free energy of the reaction and substituting it into  the thermodynamic
term of the reaction rate calculation as implemented in CrunchTope (Steefel et al., 2015) gives
Equation (4.3) below describing the rate of microbial sulfate reduction in the Rifle RTM.
$$R_{MB} = k_{max}B\frac{[\mathrm{C_2H_3O_2^-}]}{[\mathrm{C_2H_3O_2^-}]+K_{half[\mathrm{Ace}]}}\frac{\left[\mathrm{SO_4^{2-}}\right]}{\left[\mathrm{SO_4^{2-}}\right]+K_{half\left[\mathrm{SO_4^{2-}}\right]}}F_T \tag{4.3}$$

where
$$F_T = \left(1 - \frac{[\mathrm{CO_{2(aq)}}]^{\frac{1}{4}}[\mathrm{H_2S_{(aq)}}]^{\frac{1}{8}}}{[\mathrm{SO_4^{2-}}]^{\frac{1}{8}}[\mathrm{C_2H_3O_2^-}]^{\frac{1}{8}}[\mathrm{H^+}]^{\frac{3}{8}}}\right) \tag{4.4}$$

$R_{MB}$ is the overall rate of microbial sulfate reduction, $k_{max}$ the rate constant for microbial sulfate
reduction, $B$ is the biomass concentration, and $K_{half[\mathrm{X}]}$ is a half-saturation constant. The two Monod
kinetic factors for the electron donor ($\mathrm{C_2H_3O_2^-}$) and the electron acceptor ($\mathrm{SO_4^{2-}}$) are referred to as $F_D$
and $F_A$ respectively (Jin and Bethke, 2003, 2005, 2007). Equation (4.4) illustrates the underlying
relationship between pCO$_2$ in the injectate solution and the resulting accumulation of pyrite. As pCO$_2$
of in the injectate increases, the $F_T$ term becomes smaller, inhibiting the overall rate of microbial





sulfate reduction (Fig. S5). Consequently, biomass growth is also inhibited, and the rate of microbial
sulfate reduction is never high enough to produce the concentration of $H_2S_{(aq)}$ required for significant
pyrite precipitation. This explains why the model suggests that the gradient of the pyrite volume
precipitated with respect to both $C_2H_3O_2^-$ and $SO_4^{2-}$ varies as a function of $pCO_2$ in the injectate.
When $pCO_2$ is low and both $SO_4^{2-}$ and are large with respect to their half saturation constants
(Equation (4.4)), the overall Monod-biomass rate law will approach $Bk_{max}$.

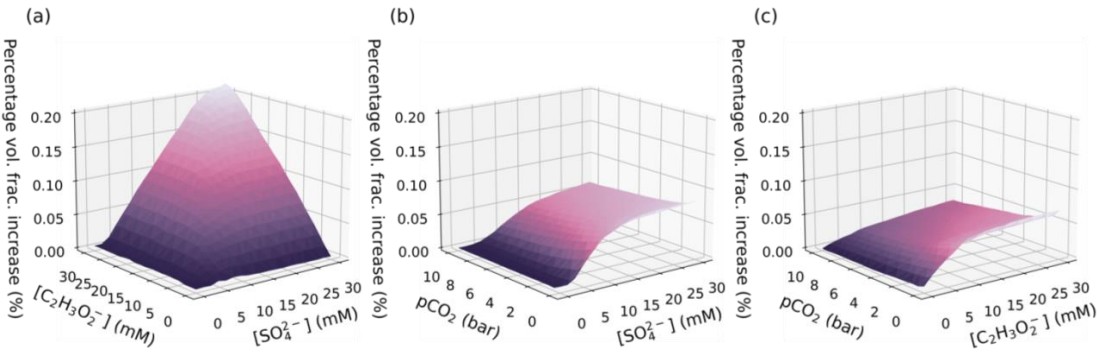


**Figure 4: A selection of the GBT model predictions of the percentage volume fraction increase**

**due to pyrite precipitation as a result of varying two parameters simultaneously for selected**
**pairs of variables. Other model parameters are held at the values used in Druhan et al. (2014).**
**The remaining variable-pair plots are provided in Fig. S4.**
This dependence emerged somewhat unexpectedly from the emulator, as one would not inherently
expect a relationship between injectate $pCO_2$ and $SO_4^{2-}$ reduction rates, yet it agrees with results
previously reported by Jin and Kirk (2016, 2018) as well as Paper et al. (2021). These studies related
the influence of $pCO_2$ and pH to the rate of microbial reactions, both *in vitro*, *in situ*, and *in silico*.
We suggest that our type of analysis could be used to direct future lab and field work to test
hypotheses suggested by the results generated by running the emulator.



This analysis also explains some of the features observed in Fig. 4A: the gradients of $C_2H_3O_2^-$ and
$SO_4^{2-}$ are coupled in such a way as to indicate that if one is in excess, then the other becomes limiting
in the production of $H_2S_{(aq)}$ and hence the precipitation of pyrite. However, the limiting behaviour
when both are in excess seems to indicate that given enough $SO_4^{2-}$ and $C_2H_3O_2^-$, pyrite precipitation
can continue indefinitely assuming suitably low pCO2. Given this prediction, it is sensible to check
whether, at such high levels of $SO_4^{2-}$ and $C_2H_3O_2^-$ as the model suggests for maximum pyrite
precipitation, there is indeed enough Fe(II) available in the system to precipitate pyrite: this is a
second potential dependence as mentioned above.
Lastly, the model can be interrogated in all 5 dimensions and the amendment fluid composition that
corresponds to the largest net pyrite precipitation over the modelled interval can be determined. This
amendment composition is shown in Table S1. The total change in volume fraction due to pyrite
precipitation predicted by the emulator is 0.143 and the actual RTM modelled precipitation when this
boundary condition is used is 0.150. There is a 4.7% absolute error on the net pyrite volume fractions
change predicted by the emulator when compared to the actual net pyrite precipitation calculated by
the RTM. This error is inherent in statistical learning techniques but can be further mitigated with
larger training datasets, in conjunction with different emulator training hyperparameterisations: an
area for future improvement to the methodology. These optimised conditions represent an almost
four-fold increase in the amount of pyrite precipitated in the original RTM for Old Rifle (Druhan et
al., 2014).
**4.2   Advantages and drawbacks of the emulation approach**
In this study, 9416 individual RTM simulations were used to train a GBT regression model to predict
a specific model output, in this case net pyrite precipitation. This emulator is a reduced representation
of the complex system of equations in the underlying RTM, having a faster computational time but





introducing some prediction errors. We now discuss the key advantages and drawbacks of this
emulation approach.

### 4.3    Advantages of the emulation approach

9416 RTM runs were used to train the emulator (the data shown in Fig. 2). This number of runs could
instead be used to perform a sensitivity analysis in all five variables at a spacing of ~4.8 mM between
points by directly interrogating the simulator. What then, is the advantage of the emulation approach,
if the same information can be visualised from discreet runs of the original RTM without having to
go to the extra effort to train the model, which introduces prediction errors? The key advantages are
outlined below.

### 4.3.1 Advantages over directly interrogating the simulator

The first and most obvious advantage is the lack of a need for an explicit interpolation scheme.
Correlations generated by directly plotting simulator results in both test cases lead to data points
lying on a grid of finite resolution. If intermediate values on this grid were to be determined, an
explicit interpolation scheme would have to be applied, which would introduce errors of its own that
would then need to be quantified. Furthermore, an improvement in the interpolation scheme would
come at the expense of adding one extra point to the grid in each dimension: in the context of Old
Rifle this is an extra 9031 data points ($7^5 - 6^5 = 9031$ going from a 5D grid of 6 points in all
directions to 7) roughly doubling the dataset size. In contrast, since any number of points can be
submitted to the emulator for inference, concerns relating dataset size to sampling resolution are
assuaged. Beyond that, the errors in the model fit are already quantified during training.
More broadly, to explore the dataspace, emulators are extremely fast compared to simulators. The
time for a single query of the emulator is on the order of milliseconds rather than the
seconds/minutes/hours for a single forward RTM simulation. This allows the emulator to be used as a





tool for efficiently exploring the simulator by rapidly developing intuition for the space itself and
how the simulator behaves in different circumstances. Furthermore, emulator models are easy to
distribute and share with collaborators. Model weights can be published directly or distributed as
standalone files. This means that a well-trained emulator can be made once and then the encoded data
shared.
Lastly, performing a direct interrogation of the simulator requires choices of parameters and ranges,
and results in a grid of points over the region of interest at limited resolution. A similar procedure
must be undertaken when creating a dataset to train the emulator, in so far as ranges and parameters
of interest must be chosen. However, the dataset can always be further added to in a straightforward
manner, further drawing from the random distribution to increase the size of the dataset and thus
improve model performance. With both approaches, using Omphalos means that the data generation
process can be parallelised and using high-performance computing facilities can reduce the
computational expense of interrogating the simulator. This means that all the computational expense
is upfront in both cases since the emulator need only be fit once.
The advantages we outline make the case for the emulator as a tool to be used in conjunction with the
RTM, rather than a replacement for it. The alacrity with which the emulator can be interrogated
means that it is an invaluable tool for investigating RTM behaviour in multiple dimensions. Further
to this, the ability to evaluate the state of a system after a fixed period of time makes the emulator
approach ideally suited for modelling more complex time-series models with time varying boundary
conditions: instead of having to run the RTM forward each time the system changes boundary
conditions, the emulator can be interrogated for the expected result given the systems current state
from the previous regime.

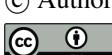



Emulation makes sensitivity analysis for RTMs simple and allows us to identify correlations and
interactions among parameters that would otherwise be difficult to anticipate, for example the $CO_2$
dependency of microbially mediated reactions (Bethke et al., 2011; Jin and Kirk, 2016, 2018; Paper
et al., 2021). This ability to elucidate unexpected but key model dependencies and sensitivities could
prove invaluable in helping direct RTM development.
**4.3.2 Application to Bayesian optimization**
A critical advantage of the technique proposed here is that working emulators are essential to
Bayesian optimization. Bayesian optimisation is an approach for finding global maxima and minima
in systems whose objective function is expensive to evaluate and does not return the gradients of that
function (of which RTMs are an example) (Frazier, 2018). Bayesian optimisation works by applying
an acquisition function that calculates the point that will give the most information about the function
that requires optimisation. An emulator is then fit using these data points selected by the acquisition
function and the emulator is updated with a new point each iteration. In this way, the optimiser
balances exploitation of known optima, and exploration of unevaluated regions of the function. Such
an approach can find the global maximum with relatively few evaluations of the RTM.
This study lays the groundwork for future application of Bayesian optimization to highly
dimensioned RTMs, potentially allowing for effective optimization over many different (twenty or
more) parameters at once. By demonstrating that broad (but local) fits to the RTM with an emulator
are possible, we have demonstrated that a GBT regressor can be used as an emulator informing a
Bayesian optimization algorithm in this context. This allows for a constellation of local fits in a
highly dimensioned space as the algorithm searches for the global optimum in problems that would
otherwise be computationally intractable. Bayesian optimisation could even be applied, with a





suitable loss function, to optimise for multiple objectives at once (subject to trade-offs among
objectives).

### 4.4    Disadvantages of the emulation approach

This emulation approach relies on the relative computational inexpensiveness of the RTM. In
situations where the underlying model is expensive or time-consuming to evaluate, and
computational resources are limited, then this modelling approach becomes unfeasible. One way to
overcome this limitation is to reduce the resolution of the RTM (as was done in this work), both in
time and space, to lower computation time but this comes at the expense of RTM accuracy. In the
context of analysing the interaction of underlying modelled processes in an RTM, this loss of
resolution may be less of a problem, as we would be primarily concerned with the relationship
among parameters and their impact on outputs, rather than their magnitudes. However, this issue of
computational expense is primarily allayed by the parallelised generation of data alluded to earlier
and only the most expensive RTMs would be intractable for a full emulator fit if this technique was
deployed correctly, and even in this extreme case, Bayesian optimisation would still be possible.
Additionally, caution is needed when choosing the ranges over which the parameters will be drawn
from the uniform random distributions. Key considerations include the number of points being
generated relative to the size of the space being covered—a denser cluster of training data will result
in a tighter fit, at the expense of range. Conversely, too small of a range and the emulator will not
capture key behaviour, or be unable to learn about simulator edge cases, as discussed above with
respect to the bioavailable iron in the Old Rifle RTM.

### 4.5    Choice of learning algorithm

Gradient boosted trees outperformed other machine-learning methods that we tested while building
the emulators, such as Gaussian process regression. The downsides of GBT include the lack of ability

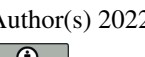



to encode smoothness to preclude sharp discontinuities in the concentration-precipitation space or
other such prior assumptions. Furthermore, a low root mean squared error over the entire model fit
region does not necessarily imply a good fit globally; it may be that there are some regions of good
fit and other regions of poor fit which make up an acceptable root mean square error over the whole
space.
**4.6    The effect of scale on emulator predictions**
Our case study relies on the capacity of CrunchTope to predict changes in mineral volume fraction.
Therefore, the errors in the predictions, and hence the utility of the approach, ultimately depend on
the scale of the system being modelled and thus the sensitivity to what could be very small changes
in mineral volume fraction.
When analysing the emulator to investigate how different processes in the underlying RTM affect
each other, we are primarily considering an issue of whether the emulator can correctly learn the
underlying model behaviour. We are also considering whether the emulator can capture the
behaviour in the output variables with respect to a changing subset of RTM parameters (some of
which we may not have expected at the outset). In this use-case, the emulator is largely concerned
with trends and gradients; Figs. Figure 3, Figure 4, S4, S8, and S9 show that this is accurately
reported in all case studies. Comparing the case study considered in this paper to the additional case
study presented in the Supporting Information we see that they are discretised at different scales (2 m
and 1 cm for the deep-sea sediment column and Old Rifle respectively). However, the emulator for
each RTM has root mean squared error over the dataset (and hence absolute error in prediction) of
the same order of magnitude. This implies that the error in absolute volume precipitated that each
model predicts is different. However, the analysis of the trends and interactions emerging from both
RTMs is equally valid in both cases.





When concerned with the optimisation capabilities of the emulator, the absolute value of the
optimised quantity and hence the model scale must be considered. In large-scale systems, such as
weathering of the critical zone, the error in the volume fraction change ($5.5 \times 10^{-5}$ for pyrite) is below
the resolution of measurement techniques for mineral abundance (e.g. XRD and SEM—(Gu et al.,
2020)). However, in smaller-scale systems where the microscale environment becomes increasingly
important, these errors in volume fraction become much harder to ignore. For example, in the RTM
experiments exploring the effects of scale on simulating mineral dissolution in porous media
described by Jung and Navarre-Sitchler (2018), significant errors in changes in predicted volume
fraction would propagate into calculated dissolution/precipitation rates, losing sensitivity in the
results.

## 4.7   Extension to multiple outputs

Multiple output regression (the prediction of a vector of outputs, rather than a single label) is in
active development for XGBoost and is currently available for other machine learning
implementations that we explored, including GPFlow for Gaussian process regression. Given that our
approach is currently limited to the prediction of one label-per-emulator trained, the availability of
regressors that can predict more than one label 'off the shelf' will greatly improve the utility of
reactive transport emulation. The prediction of multiple outputs simultaneously will expand the scope
of analysis to investigate the interaction of modelled processes in multiple outputs at once. In the
context of optimisation problems, one possible application of the emulator like this could be to
maximise mineral precipitation in one region of a system while trying to maximise dissolution in
another region.

## 4.8   Improvements to the model





This proof-of-concept model demonstrates the fitting of an emulator over a relatively small range of
environmental parameters. Future work will involve expanding the scope of the emulators both in
terms of the number of parameters being varied, but also the range over which they are varied, so the
whole behaviour of the underlying model can be captured with more accuracy. There is also scope
for adding time dependency to the GBT modelling approach, to predict a time series intermediate
RTM states during the evolution of geochemical systems.
**4.9    Potential applications**
Our emulator approach is flexible; any quantity recorded by an RTM can be used as a target variable,
and so the behaviour of any RTM output can be explored in detail to evaluate the model formulation.
The behaviour of the system in response to the variation of any parameter under any other set of
conditions can be projected out of the model and plotted in a straight-forward manner. This approach
can be extended to two or even three dimensions and time series thereof and ultimately the emulator
can be interrogated for local maxima and minima to solve optimisation problems. This new approach
has potential applications in industry and in environmental remediation where the chemical
composition of amendments can be predicted using an underlying reactive transport simulation,
provided that that system is well understood.
Omphalos also has utility outside of generating datasets for emulation; its automated submission of
CrunchTope input files means it can be used to systematically explore sets of input variables in an
easy way, simply by editing the Omphalos configuration file.
**5    Conclusions**
We have presented a new approach for interrogating and understanding multi-component RTMs. By
building an emulator of an RTM that captures the multidimensional nature of the underlying model
we have created a new tool for performing global sensitivity analyses on RTMs. This allows us to





investigate behaviour arising from the interaction among the many disparate processes that comprise
RTMs. For example, we investigated how the Monod-biomass parameterisation of microbial sulfate
reduction interacted with the mechanism of pyrite precipitation. In this example, pyrite precipitation
was inhibited when there was an excess of $CO_2$ in the column because the catabolic pathway was
partially dependent on $CO_2$ concentration. This prevented the growth of sulfate reducing biomass,
ultimately curtailing the production of hydrogen sulfide required for pyrite precipitation. This
behaviour reproduced results previously reported by Jin and Kirk (2016, 2018), and suggest that our
emulation approach has utility in discovering unexpected, but nonetheless real, model behaviours,
potentially directing future lab and field work.
The approach is flexible; any quantity recorded by an RTM can be used as a target variable, and so
the behaviour of any RTM output can be explored in detail to evaluate the model formulation. The
behaviour of the system in response to the variation of any parameter under any other set of
conditions can be projected out of the model and plot in a straight-forward manner. This approach
can be extended to two or even three dimensions and ultimately the emulator can be interrogated for
local maxima and minima to solve optimisation problems. This new approach has potential
applications in industry and in environmental remediation where the chemical composition of
amendments can be predicted using an underlying reactive transport simulation, provided that that
system is well understood. The presentation of this optimisation process to Old Rifle (and to ODP
Site 1086, see supplementary) represents a proof of concept.
**6    Code availability**
**6.1    Omphalos**
Omphalos is available on GitHub and Zenodo. Please note you must provide your own CrunchTope
executable.



https://github.com/a-fotherby/Omphalos
https://doi.org/10.5281/zenodo.7113298
**6.2    GBT Models**
Jupyter notebooks for fitting the GBT models and plotting the figures are available on GitHub, and a
permanent record is available on Zenodo.
https://github.com/a-fotherby/dissertation_xgboost.
https://doi.org/10.5281/zenodo.7113323
**7    Data availability**
The data used is available on GitHub and Zenodo.
https://github.com/a-fotherby/GMD_2022
https://doi.org/10.5281/zenodo.7113379
**8    Supplement**
Codebase for Omphalos. Model fitting code. Schematic figures of decision tree and the Old Rifle
RTM. Table of predicted optimal values for precipitating pyrite at Old Rifle. Convergence behaviour
of the GBT regressors. Additional co-dependency plots for Old Rifle. Figure showing the effect of
rate law choice on $CO_2$ dependency in the Old Rifle RTM. Supplementary Case Study detailing
application to a deep-sea sediment column. Description of XGBoost implementation.
**9    Author contribution**



AF and HJB conceived of the study. AF wrote the codebase and conducted the experiments. AF
prepared the manuscript with contributions from all co-authors.
**10    Competing interests**
The authors declare that they have no conflict of interest.
**11    Acknowledgements**
The work was supported by NERC NE/R013519/1 to HJB and by a call for International Emerging
Actions granted by the CNRS (TELEMAART: Trace ELEments and inverse Models: Advancing
Applications of Reactive Transport models) to JLD. This work was also funded by
ICA\R1\1801227 from the Royal Society to AVT.

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
