# Peer review of "An emulation-based approach for interrogating reactive transport models"

_EGUsphere, 2022_

## Author Response (AR1)

**Omphalos paper GMD submission review round 1 responses**

**Reviewer 1**

We thank the reviewer very much for taking the time to read and review our manuscript, as well as for their insightful questions and comments. The reviewer is correct in pointing out that we restrict ourselves to emulating 1D reactive transport models only. The reason for this is two-fold: first this manuscript is primarily designed as a proof-of-concept for the technique, and as such we wanted to apply it to previously published reactive transport models that had a high degree of geochemical complexity but were not so focussed on the transport aspect of the system for the method development. Second, we wanted to apply our emulator technique to solve a simple geochemical optimisation problem for each system (Section 4), to show that the approach had applicability beyond preforming sensitivity analysis, and we feel this capability is best demonstrated in a 1D system with geochemical complexity. We do acknowledge that transport is an important part of geochemical modelling, and that transport is not typically in 1D, and this represents a future challenge for emulator development in this space, but we would suggest that the primary focus here is on geochemistry and not transport.

The reviewer suggests that we should expand our scope to include 2 or 3D models lest our approach be "just another way of doing sensitivity analysis". We appreciate this comment but suggest that it is for future work. All emulators of reactive transport models, regardless of whether they deal with transport in 1, 2 or 3 dimensions, are extremely well equipped for doing sensitivity analysis. In the revised manuscript, we acknowledge this as a potential (and useful) aspect of the approach we present (e.g. line 459) but we also demonstrate the ability of the emulator to solve simple geochemical optimisation problems in two very different geochemical systems (Section 4, and Supplementary Section 3.2), which is a future direction in which we hope to take this research, among other things.

The reviewer does however erroneously suggest that the manuscript rehashes work previously done at Old Rifle by one of the co-authors. This is not the case; this work has not been conducted before. This is the first time an emulator has been developed to reproduce a previously published RTM for Old Rifle (line 135).

We now turn to the reviewer's specific comments in turn.

1. The reviewer queries our statement that "we use the emulator to explore how varying the boundary conditions in the RTM describing the aquifer impacts the rates and volumes of mineral precipitation." (lines 17–19). This is true, the new boundary conditions were used as labels for the net pyrite precipitated in the column (see section 3.4.1. Data Strategy, lines 230–232 and 239–243). The emulator must be trained to emulate the system based on some dataset but the reviewer is right that an interpolation scheme could be used—there is a full discussion of the advantages of emulation over interpolation in Section 4.3.1. The reviewer is also right to point out the potential hazards of emulators extrapolating beyond the trained region. Although we do not do this in this manuscript, there is a brief discussion of such extrapolation in lines 321–331.

2. The reviewer also asks about our discovery of an unanticipated dependence of pyrite precipitation on $pCO_2$ and how it is that an emulator, which is necessarily a reduced-order model, can provide more insight into the underlying RTM. This is a result of the growing complexity and sophistication of modern RTMs, which is one of our motivations in developing this methodology. Modern RTMs draw large suites of chemical and mineralogical data from vast databases, which constitute large sets of non-linear equations all coupled through transport and fluid chemistry—it is inevitable that in their development there will be feedbacks between quantities that are not realised. There is nothing inherent about reduced dimensionality that prevents such feedbacks from being captured in a dataset and learned by an emulator. In fact, we suggest that emulators are remarkably well placed for exploring such feedbacks because of the speed at which they can be interrogated. We acknowledge in lines 389–394 that such feedbacks need to be (subsequently) tested in the field and lab but suggest that the use of emulators in this way could be an interesting way to direct future research.

3. The reviewer asks if we validated our finding that pyrite precipitation has a dependency on $pCO_2$ using a reactive transport model. The results of us exploring and verifying this dependency in CrunchTope are shown in Figure 3 of the revised manuscript. We also discuss the mechanism by which this dependency occurs in lines 350—383.

4. Reactive transport models are forward simulations built on a mechanistic understanding of the geochemical systems that they attempt to model. As such they lack built-in capability for finding the conditions under which they might maximise a given geochemical quantity. On the other hand, emulators are reduced representations of RTMs that synthesise a lot of data about the underlying model and can be run very quickly, making them ideally placed for finding maxima and minima. In the model, we do this by interrogating the emulator at regular intervals to find an approximation of the maxima. The reviewer is correct in pointing out that this is not made explicit, and we clarify this in the revised manuscript (lines 404–409). This approach is a simple one, but we do discuss more sophisticated, future applications of emulation techniques (Bayesian Optimisation) for doing this as well in section 4.3.3.

5. The reviewer correctly points out that our claim that RTMs are computationally expensive does seem to be at odds with the large data set used to train the emulator and suggests that emulators of 2 or 3D RTMs would enhance the manuscript. We would suggest that modelling 1D flow is no guarantee of computational speed and that our focus in this manuscript is more focussed on the geochemistry than the transport (see our overarching comment above).

6. The reviewer asks how it is that we ensure that our synthetic data is realistic. This is an excellent point and something we clarify in the revised manuscript. The data that was excluded was due to runs failing to complete before an arbitrary cut off time, or due to extreme boundary conditions that failed to

speciate. This is now clarified in the revised manuscript, lines 194–199. We ensure synthetic data is realistic by having a well validated underlying RTM with a sound set of physical processes governing the behaviour. The developer of an RTM will know what assumptions have gone into their model and hence know where it is valid to probe with the emulator and where it is not.

7.  The reviewer asks about the data shown in Figures 3A and 3C and whether the emulator was exposed to data about the specific boundary conditions represented by the black crosses. The answer is no, they have not been exposed to those exact conditions, as the data for training was generated by a random sample (see Section 3.1) and we clarify this in the revised manuscript (lines 301–303). This is now in the figure caption to clarify. The reviewer also suggests that the fit lines in blue do not capture the trends shown in the underlying black data in Figures 3A and 3C. It is true that these fits are slightly offset but in both cases the error is small and does not greatly impact the conclusions we draw. We suggest that the reason why this may be is that weak/non dependencies get swamped by the larger signals in the dataset and are thus slightly drawn down on average. We thank the reviewer for the suggestion the manuscript would benefit from a discussion of this point and have implemented it (lines 306–310).

8.  The reviewer is alluding to testing the emulator on unseen data here. This has been done for both models and is shown in Figure 3 and Figure S8. We have also clarified our validation strategy and provided validation testing scores in the revised manuscript, lines 269–277.

We thank the reviewer for their time and help in improving this manuscript.

**Reviewer 2**

We thank the reviewer for taking the time to read and assess our manuscript. We would immediately point out that the two reactive transport models that we emulate in this paper are 1D models (e.g., see line 215). There are figures that make this clearer in the supplementary and perhaps the paper would benefit from returning those to the main manuscript body, but we have added a clarifying line earlier in the manuscript (line 98). Otherwise, we thank the reviewer for their concise and accurate overview. We will now respond to the reviewer's specific concerns in turn:

1.  The reviewer correctly points out that the novelty of this paper is now a lot weaker than when it was first conceived back in 2020 and initially reviewed early 2021, as emulator approaches have now been applied in the RTM context in a variety of ways. We apologise for missing the papers suggested when this manuscript was posted to the GMD pre-print server nearly a year ago, which demonstrate how emulators can be used to speed up RTM simulations by replacing the geochemical solver with an emulator. Our is approach is closer in nature to the 2$^{nd}$ paper suggested to us (Ahmmed et al., 2021), which tests the ability of a wide variety of machine learning approaches to predicts the degree of mixing and production of a hypothetical species C

from two reactants A and B. We extend this underlying principle to RTMs of real-world systems to develop new ways to explore geochemical parameters spaces and the effect of changing those geochemical parameters on the overall outcome of reactive transport simulations, with an eye towards predicting system outcomes in real world scenarios. We have added a paragraph to this effect in the revised manuscript (lines 74–85). We have removed references to the novelty of this approach in the revised manuscript, as it is no longer the case and have included references to the suggested papers. It is unfortunate that this paper spent nearly two years in review, so we are not the first to show this approach.

2. The reviewer suggests that the motivation to build emulators is weak in the original submission. Leaving aside the point that the models we are emulating are 1D, rather than 0D models, we suggest that emulators have utility in their ability to enable optimisation routines (Section 4.3.3, for example) for RTMs, as well as their ability to facilitate the exploration of the geochemical space in an efficient way, which can allow for the discovery of new feedbacks (which the reviewer touches on in their next point) as well as performing efficient sensitivity analysis. We address these points in Sections 4.2 and 4.3 but we have added a comment clarifying that we are primarily interested in exploring the highly dimensioned geochemical space (lines 96–98). We thank the reviewer for the suggestion.

3. The reviewer indicates that a careful discussion of how the emulator approach can be used to gain new insights into the geochemical behaviour of RTMs is needed. Of course, in a sufficiently simple model, coupled geochemical behaviour can be deduced by reasoning about the governing equations. However, modern RTMs draw large suites of chemical and mineralogical data from vast databases, which constitute large sets of non-linear equations all coupled through transport and fluid chemistry—it is inevitable that in their development there will be feedbacks between quantities that are overlooked. The reduced representation of the emulator allows investigators to quickly test a large variety of different hypotheses. Ultimately, we suggest that the benefit comes in providing another avenue for discovery and investigation and this is borne out by the fact that the original RTM for the Old Rifle was published in 2012, but the effect of $pCO_2$ on pyrite precipitation was not reported until 2016 (see line 351). We agree that this point needs to be clarified and have added an additional subsection discussing this in the under section 4.3.

4. The reviewer rightly points out that there is a lack of structured discission of the testing and training metrics for our emulators in the main body of the manuscript, although there is some in the supplementary, e.g. Figure S2. We have updated the manuscript to include our emulator testing and training results, lines 269—277.

We thank the reviewer for giving their time and effort to improve this manuscript.